# Rethinking Sampling in 3D Point Cloud Generative Adversarial Networks

## Abstract

In this paper, we examine the long-neglected yet important effects of point sampling patterns in point cloud GANs. Through extensive experiments, we show that sampling-insensitive discriminators (*e.g.* PointNet-Max) produce shape point clouds with point clustering artifacts while sampling-oversensitive discriminators (*e.g.* PointNet++, DGCNN, PointConv, KPConv) fail to guide valid shape generation. We propose the concept of sampling spectrum to depict the different sampling sensitivities of discriminators. We further study how different evaluation metrics weigh the sampling pattern against the geometry and propose several perceptual metrics forming a sampling spectrum of metrics. Guided by the proposed sampling spectrum, we discover a middle-point sampling-aware baseline discriminator, PointNet-Mix, which improves all existing point cloud generators by a large margin on sampling-related metrics. We point out that, given that recent research has been focused on the generator design, the discriminator design needs more attention. Our work provides both suggestions and tools for building future discriminators. We will release the code to facilitate future research.

## 1 Introduction

Point cloud, as the most common form of 3D sensor data, has been widely used in a variety of 3D vision applications due to its compact yet expressive nature and its amenability to geometric manipulations. It is natural to consider how to generate point cloud through deep learning approaches, which has been a popular research topic recently. The previous research efforts in the community have been mainly devoted to conditional generation of point clouds with 3D supervision. The condition could either be images (Fan et al., 2017; Groueix et al., 2018; Park et al., 2019) or partial point clouds (Li et al., 2019; Yang et al., 2018).

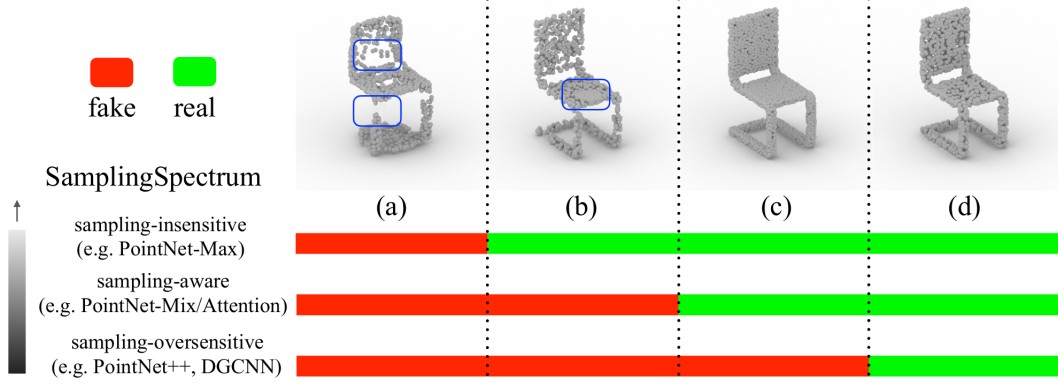

Figure 1: We visualize the behavior of different discriminators when judging four different chair point clouds. (a) and (b) are generated results, (c) and (d) are point clouds sampled using FPS and uniform sampling from a real chair surface. When training the discriminators on data like (d), different discriminators make distinct decisions on the point cloud realness forming a **sampling spectrum** ranging from sampling-insensitive, sampling-aware to sampling-oversensitive. We advocate sampling-aware discriminators in the middle of the spectrum, which provide good guidance for fixing geometric flaws (a) and big sample artifacts (b) and tolerate subtle sampling differences between (c) and (d).

Generating 3D point clouds with GANs in an unsupervised manner is an important but less explored problem. 3D point cloud GAN learns to transform a random latent code into a 3D surface point cloud by playing an adversarial game. Its development is still in an early stage compared with 2D image GANs. While existing works such as (Achlioptas et al., 2018; Valsesia et al., 2018; Shu et al., 2019) have developed a variety of generators, they all use PointNet (Qi et al., 2017a) with max pooling (PointNet-Max) as their discriminator. PointNet, which is essentially a pointwise MLP followed by a global pooling operation, is too limited in capturing shape details for a successful GAN. However, advanced networks, e.g. PointNet++(Qi et al., 2017b), DGCNN(Wang et al., 2019), KPConv (Thomas et al., 2019), PointConv (Wu et al., 2019), which leverage relative positions between points and hierarchical feature extraction, may not help. From our empirical study, we find they all fail to be a functioning discriminator. Understanding their failure mechanism and improving discriminator design are hence important and urgent.

To design a better discriminator, we first need to answer the following question: *what should the discriminator examine for improving the generation quality?* Or, even more fundamentally, *what does it mean by the quality of generated point clouds?*

Since a shape point cloud are the points sampled from an object surface, its quality should be evaluated from two perspectives: *the depicted surface geometry* and *the point sampling*. Arguably, geometry plays a decisive role and should be the main focus of a discriminator. However, when the generated point clouds have a good shape, there is still a full spectrum on how much a discriminator cares about the sampling patterns. We introduce the concept of **Sampling Spectrum** to depict the *sampling sensitivity of discriminators*, as illustrated in Figure 1. A sampling-insensitive discriminator (*e.g.* PointNet-Max) may ignore the point density variations as long as it perceives a good overall shape. Such a discriminator could identify big geometric flaws as shown in Figure 1 (a), but turns a blind eye to highly non-uniform density distribution, *e.g.* point clusters in Figure 1 (b). On the other extreme, a sampling-oversensitive discriminator (*e.g.* PointNet++, DGCNN, KPConv) can even tell the subtle difference in sampling patterns, *e.g.* between furthest point sampling (FPS) in Figure 1 (c) and uniform sampling in Figure 1 (d), and hence can be very narrow-minded about what a real point cloud should look like. A sampling-aware discriminator, *e.g.* PointNet-Mix/Attention, which lies in the middle of the spectrum, is able to identify density-related artifacts such as point clusters in Figure 1 (b), while not being too sensitive to different sampling patterns, such as Figure 1 (c) and (d).

Resembling the sampling spectrum of discriminators, we also examine the existing point-cloud GAN evaluation metrics from the perspective of sampling sensitivity and propose several perceptual metrics forming a **Sampling Spectrum** of *evaluation metrics*. Understanding how the metrics weigh between sampling and geometry is a prerequisite for evaluating point cloud GANs. Many of the existing sampling-insensitive metrics only evaluate the geometry factor of the generated point clouds shapes, which are blind to the obvious point clustering artifacts and uneven point density. We propose novel sampling-sensitive metrics to further complete the spectrum of point-cloud GAN evaluation metrics.

Guided by the proposed sampling spectrum of discriminators and evaluation metrics, experiments show that different discriminators in the spectrum could provide very different suggestions to improve a generator according to its sampling sensitivity. Sampling-insensitive discriminators, *e.g.* PointNet-Max, is unaware of point density variations and hence its generated point cloud inevitably suffer from clustering artifacts, while sampling-oversensitive discriminators, *e.g.* PointNet++, DGCNN, KPConv, PointConv, simply fail to function as the discriminators and can generate much degraded point cloud shapes. We design a diagnostic "no generator" experiment to factor out the impact from generators and reveal that the gradients of sampling-oversensitive discriminators prioritize adjusting sampling patterns over producing better shape geometry. Picking a middle-point on the sampling spectrum, we discover a simple yet effective sampling-aware discriminator, PointNet-Mix, and find that it can supervise both shape generation and point density uniformity. It improves all existing generators by a large margin on sampling-related metrics. Surprisingly, we find that even the most naive fully-connected generator, coupled with PointNet-Mix, simply beats all the start-of-the-art point cloud GANs. This discovery conveys an important message to the community: instead of focusing on the generator design, people should invest more time into discriminator and seek for more powerful sampling-aware discriminators.

## 2 POINT CLOUD GAN LANDSCAPE

In this section, we review the current state of point cloud GAN covering the generators, the discriminators, and the evaluation metrics we are examining in this work.

## 2.1 POINT CLOUD GAN GENERATORS

Recent point cloud GAN works primarily focus on the generator design. The generator takes input a random noise $z \sim \mathcal{N}(0, I)$ and outputs a point cloud $p \in \mathbb{R}^{N \times 3}$. The existing generators can be categorized into two classes: fully-connected (FC) generators and graph convolutional generators. The first point cloud GAN, r-GAN (Achlioptas et al., 2018), simply uses an FC network as its generator. GraphCNN-GAN (Valsesia et al., 2018) and TreeGAN (Shu et al., 2019) are the rest two published works in this field that use graph convolutional generators. The two methods are very similar in principle. The main difference lies in how they build graphs. GraphCNN-GAN builds a dynamic $k$-nn graph based upon feature space distance while TreeGAN enforces a tree structure throughout its sequential graph expansion and the messages can only be passed from ancestors vertices to descendants vertices.

Deformation-based decoders (Groueix et al., 2018; Yang et al., 2018) are widely used in the point cloud auto-encoder networks for 3D shape reconstruction. The decoders leverage Multiple-layer Perceptrons (MLP) to deform template surfaces into shape surfaces taking as inputs the concatenation of template point coordinates and the latent shape feature vectors. Though the decoders can truly act as generators for point cloud GANs, they have not yet been used in unconditioned point cloud GAN literature. Recently, Mo et al. (Mo et al., 2020) use deformation-based decoder as part generators for structure-conditioned point cloud GAN.

## 2.2 POINT CLOUD GAN DISCRIMINATORS

All existing works on unconditioned point cloud GANs use PointNet with max-pooling (PointNet-Max) as their discriminators. The discriminator takes input a point cloud $p \in \mathbb{R}^{N \times 3}$ and outputs a score from 0 to 1. PointNet learns a function $h$ that maps each point $p_i$ in the point cloud to a per-point feature $h(p_i) \in \mathbb{R}^d$ and then extracts a permutation-invariant global feature $F \in \mathbb{R}^d$ by pooling the per-point features across all points using a symmetric function $g$, which can be max pooling, average pooling, etc. Namely, we have $F = g(\{h(p_1), h(p_2), \cdots, h(p_N)\})$.

**PointNet-Max/Avg.** Though $g$ can be any symmetric function, most existing works use max-pooling. In PointNet, the authors show that max-pooling outperforms average-pooling on 3D shape classification tasks. They further show that the global feature $F$ obtained from max-pooling is determined by only a sparse subset of the points, namely critical points $\mathcal{C}_S$, bringing PointNet-Max with robustness against small data perturbation and corruption. However, this property may limit its discriminative power on telling the density variations and classifying different sampling patterns. To investigate how different aggregation operations affect the sampling sensitivity and generation quality, we study the two common choices of the symmetric function $g$, max-pooling, and average-pooling, in this paper. We will show in Sec. 3 that using different aggregation operations makes huge differences when adapting PointNet-based networks as point cloud GAN discriminators.

**PointNet-Mix.** By simply concatenating the max-pooling feature and the average pool feature, we obtain another permutation-invariant feature. We name this PointNet-Mix. Formally, $F_{\text{mix}} = [\max\{h(p_1), ..., h(p_N)\}; \text{avg}\{h(p_1), ..., h(p_N)\}] \in \mathbb{R}^{2d}$. The mix-pooling operation is a special choice of the symmetric function $g$. We will show in our experiments that PointNet-Mix, though simple, surprisingly improves the performance for most point cloud GANs by a large margin on sampling-aware metrics.

**PointNet-Attention.** A recent point cloud upsampling work (Li et al., 2019) incorporates a self-attention module into PointNet for its discriminator and shows improved point density in the upsampled point cloud. We denote this discriminator as PointNet-Attention. The self-attention module learns three separate MLPs to transform each $h(p_i)$ into $f(p_i), l(p_i), k(p_i) \in \mathbb{R}^d$ correspondingly. Then an attention weight matrix is formed by $W = \text{SoftMax}\left([f(p_1), ..., f(p_N)]^T[(l(p_1), ..., l(p_N))]\right) \in \mathbb{R}^{N \times N}$. A weighted features is then obtained through $w(p_i) = h(p_i) + W^T k(p_i)$. The final aggregated features of PointNet-Attention is a max-pooling of the weighted feature, namely $F_{\text{attention}} = \max\{w(p_1), ..., w(p_N)\}$. Note that PointNet-Attention allows the per-point features to communicate with each other according to their similarity, which is more sensitive to sampling patterns than PointNet-Max. We investigate this strategy in our paper as well.

**Discriminators beyond PointNet.** Recently, there have been many works (Qi et al., 2017a; Li et al., 2018; Wang et al., 2019; Hermosilla et al., 2018; Thomas et al., 2019) extending PointNet to more advanced 3D deep learning architectures on point clouds. They improve PointNet by extracting more local or hierarchical geometric features via point cloud convolutions or point cloud graph learning. Though proven to be effective on shape classification and segmentation tasks, no published work examines adapting them as point cloud GAN discriminators. In this paper, we investigate four

exemplar beyond-PointNet discriminators: PointNet++ (Qi et al., 2017b), DGCNN (Wang et al., 2019), KPConv (Thomas et al., 2019), PointConv (Wu et al., 2019).

## 2.3 POINT CLOUD GAN EVALUATION METRICS

Achlioptas et al. (Achlioptas et al., 2018) introduces two distance metrics in the Euclidean space for evaluating point cloud GAN: the coverage scores (COV) computing the fraction of the point clouds in $B$ that are closed to the point clouds in $A$ using either Earth Mover's distance (EMD) or Chamfer distance (CD), and the Minimum matching distance (MMD) scores measuring the fidelity of $A$ with respect to $B$ using either EMD or CD. In the field of 2D image GAN, it is common to use perceptual metrics, such as the Frechét distance (Heusel et al., 2017) between real and fake Gaussian measures in the feature spaces, for evaluating the generated image results. Formally, Frechet Distance $= ||\mu_r - \mu_g||^2 + \text{Tr}(\Sigma_r + \Sigma_g - 2(\Sigma_r \Sigma_g)^{1/2})$, where $\mu$ and $\Sigma$ are the mean vector and the covariance matrix of the features calculated from either real or fake data distribution, and Tr is the matrix trace. For point clouds, Shu et al. (Shu et al., 2019) proposes Frechét Point Cloud Distance (FPD), which uses the features extracted from a pre-trained PointNet-Max model. In this paper, we position the existing metrics on a sampling spectrum according to their sampling sensitivity and propose novel sampling-aware metrics to augment the spectrum.

## 3 SAMPLING SPECTRUM OF DISCRIMINATORS

While recent works propose many advanced generator improvements for point cloud GANs, we find that designing a good discriminator is of equal importance, if not more. In this section, we introduce *the sampling spectrum of discriminators*, on which we thoroughly examine the positions of different discriminators that explain their behaviors when training point cloud GANs.

### 3.1 SAMPLING SENSITIVITY OF DISCRIMINATORS

The sampling sensitivity of a discriminator depicts how much it responds to a change in the point density or sampling pattern of an input point cloud. We find it extremely hard to quantitatively measure this sensitivity given the difficulty of measuring changes in the sampling patterns. Naively using Euclidean metrics (*e.g.* CD or EMD) to measure the distance between two sampled point sets is not a solution, since given the same distance budget, the discriminator's responses can be dramatically different depending on how the sampled points move. Instead, we can set landmarks in the continuous spectrum of sampling sensitivity by examining the discriminative power of the discriminators against different sampling patterns under a series of experiments from easy to hard. Specifically, we design two experiments to test whether a discriminator could tell clustering artifacts in point clouds and whether it could distinguish between FPS and uniform sampling patterns. Accordingly, we divide the spectrum into three regimes: sampling-insensitive/-aware/-oversensitive, as shown in Fig.1.

**A sampling-insensitive discriminator** does not respond to local point density changes if the overall shape remains roughly the same. This kind of discriminators can't tell clustering artifacts, i.e. Fig.1 (b), and thus may cause the non-uniform density in the generated point clouds.

**A sampling-aware discriminator** can notice the significant point non-uniformity and is hence capable of supervising the generator to enforce a similar sampling distribution to the training data, while being ignorant to subtle changes when the sampling is already uniform in an intermediate scale. Such discriminator will judge Fig.1 (b) as fake but can't tell the difference between Fig.1 (c) and (d).

**A sampling-oversensitive discriminator** can tell very subtle difference in the sampling patterns, *e.g.* between FPS and uniform sampling, even if the underlying shapes are the same (Fig.1 (c) and (d)).

### 3.2 SAMPLING SENSITIVITY EXAMINATION RESULTS

We design diagnostic experiments to categorize point cloud GAN discriminators on the sampling spectrum: sampling-insensitive (PointNet-Max), sampling-aware (PointNet-Avg, PointNet-Mix, PointNet-Attention (Li et al., 2019)), and sampling-oversensitive (PointNet++, DGCNN, KPConv, PointConv).

**The Discriminating Power against Clustering Artifacts.**  To examine whether the discriminators can tell clustering artifacts or not, we create a diagnostic classification dataset using the models from ShapeNet (Chang et al., 2015). Taking 100 chairs, we uniformly sample 2048 points from each shape, forming a set of real point clouds. To form a fake set of point clouds, for each chair, we first uniformly sample 1024 points, and then densely sample another 1024 points around a random position on the chair within a 0.1 radius. The real/fake point clouds are used as the labeled training dataset. We repeat the same process to generate a test dataset using a different set of 100 chairs.

| Experiment | PN-Max | PN-Avg | PN-Mix | PN-Att | PN++ | DGCNN | PointConv | KPConv |
|---|---|---|---|---|---|---|---|---|
| Clustering art. | 56%/53% | 98%/99% | 97%/96% | 94%/93% | 100%/98% | 100%/99% | 100%/99% | 100%/99% |
| FPS vs. uniform | 50%/50% | 50.3%/50% | 51.5%/50% | 50.1%/50% | 100%/97% | 100%/96% | 100%/98% | 100%/99% |
| ESS | 0.03 | 0.49 | 0.46 | 0.43 | 0.95 | 0.95 | 0.97 | 0.99 |

Table 1: Evaluating the discriminating power of the discriminators against clustering artifacts and sutble change in sampling patterns along with their empirical sampling sensitivity (ESS). In each cell of the top two rows, the left shows the training accuracy while the right shows the test accuracy.

We supervisely train each discriminator to classify the real/fake point clouds. We train them 200 epochs until convergence. The training and test accuracies are shown in the first row of Table 1. Despite the huge density variation and the remarkable clustering artifacts, PointNet-Max is just barely better than a random guess while the rest of the discriminators are very successful in telling the fake from the real. This indicates that *PointNet-Max is sampling-insensitive*. Note that the discriminating power of a network towards certain artifacts is maximized under such supervised training scheme. A network will fail to identify such artifacts in an adversarial training scheme if it fails in a supervised training scheme. We will see in Sec. 5.2, point clouds generated by GAN using PointNet-Max as the discriminator indeed suffer from non-uniform density artifacts.

Our insight why PointNet-Avg/Mix can tell the artifacts but PointNet-Max fails is that the average pooling feature computes the center of the mass of the points in feature space and is hence aware to certain global non-uniform density distributions. PointNet-Attention leverages a learnable weighted averaging and is hence capable to identify the difference.

**Distinguishing between FPS and Uniform Sampling.**  We construct another diagnostic dataset with real and fake data which are the same in their shapes but only differ in their sampling patterns. Specifically, we perform uniform sampling to generate real data and use FPS to generate fake data from 100 chairs. Figure 1 (c) and (d) illustrate the different sampling pattern outcomes.

We present the training and test accuracies in the second row of Table 1 and show that PointNet++, DGCNN, KPConv, and PointConv can perfectly distinguish the subtle difference in sampling patterns while the rest discriminators make no progress even on the training set. The experiment indicates that *PointNet-Avg/Mix/Attention are sampling-aware* while *PointNet++, DGCNN, KPConv, and PointConv are sampling-oversensitive*.

We believe, for PointNet++, DGCNN, KPConv, and PointConv, their capability of distinguishing FPS from uniform sampling owes to their usage of relative point positions or edge information, which are highly sensitive to any change in sampling. We will show in Sec. 5.3 that their remarkable discriminating power on sampling patterns actually leads to the failures as functioning discriminators to train point cloud GANs.

**Empirical Definition of Sampling Sensitivity**   Given the difficulty of measuring the change in sampling pattern, we would like to provide an empirical definition of sampling sensitivity of one discriminator according to its performance on the two experiments above: when the discriminator fails in telling clustering artifacts and telling the difference between FPS and uniform sampling pattern, its sampling sensitivity should be closed to 0; when the discriminator can tell clustering artifacts but can't distinguish between FPS and uniform sampling, we want to assign it a sampling sensitivity closed to 0.5 ; when the discriminator can do both, we want to assign a sampling sensitivity closed to 1 to it. Based on this, we define empirical sampling sensitivity $ESS \in [0, 1]$ as:

$$ESS = \text{Acc}(\text{Tell clustering art.}) + \text{Acc}(\text{Tell FPS from uniform sampling}) - 1,$$

where Acc denotes the test accuracy of an experiment. See Table 1 for results. Using this metric, we would form a continuous sampling sensitivity spectrum.

## 4    SAMPLING SPECTRUM OF EVALUATION METRICS

Similar to the discriminator design, it is very important to understand how different evaluation metrics weigh the differences in sampling patterns against geometry quality. Thus, we introduce *the sampling spectrum of evaluation metrics*, which exactly resembles the sampling spectrum of discriminators introduced in Sec. 3. On the spectrum, we have sampling-insensitive metrics that measure only the shape geometry and are ignorant of the sampling patterns, and sampling-sensitive metrics that measure both at the same time.

For perceptual metrics, *i.e.* Frechét distances in feature spaces, the sampling sensitivity of the metric purely depends on the sampling sensitivity of its feature extractor. Frechét distance measured in

| Data | FPD-Mix ↓ | FPD-Max↓ | FGD↓ | MMD-E↓ | MMD-C↓ | COV-E↑ | COV-C↑ |
|---|---|---|---|---|---|---|---|
| Uniform re-sampling | 0.1153 | 0.0926 | 0.8141 | 0.1104 | 0.00145 | 70.69 | 72.16 |
| Farthest point sampling | 0.1700 | 0.1558 | 1.8833 | 0.1064 | 0.00137 | 67.74 | 69.36 |
| Biased sampling | 2.8631 | 0.3524 | 9.6719 | 0.2469 | 0.00145 | 23.12 | 71.28 |

Table 2: Examining sampling sensitivity of evaluation metrics.

different feature spaces may respond very differently to changes in point density and sampling patterns. In this work, we examine three Frechét distance metrics that extract features using PointNet-Max, PointNet-Mix, and DGCNN, respectively. We denote them as Frechét PointNet-Max Distance (FPD-Max), Frechét PointNet-Mix Distance (FPD-Mix), and Frechét DGCNN Distance (FGD). We pretrain all the three feature extraction networks on ModelNet40 (Wu et al., 2015) shape classification.

To examine the sampling sensitivity of FPD-Mix/Max and FGD, we create several copies of the training split of our ShapeNet chair dataset (see Sec.5.1), each of which uses a different sampling strategy to obtain the shape point clouds. The reference one used as the ground truth is using uniform sampling. Then we consider 1) uniform sampling with a different random seed; 2) FPS; and 3) biased sampling with clustered artifacts (as described in Sec. 3.2). We use all the available metrics to evaluate their distances to the ground truth data. The results are shown in Table 2.

We observe that *the Frechét distance metrics share the same sampling sensitivity of their corresponding discriminators*. For example, since PointNet-Max is sampling-insensitive, FPD-Max remains very low even on biased sampling data, hence FPD-Max serves as a perceptual geometry metric, which is ignorant to sampling patterns. Similarly, we find that FPD-Mix is sampling-aware since it clearly detects the biased sampling patterns while not being able to distinguish the uniform sampling and FPS, while FGD is sampling-oversensitive in that it can tell apart FPS and uniform sampling.

For Euclidean distance metrics, results in Table 2 show that COV-EMD and MMD-EMD are sampling-aware, which is intuitively reasonable since EMD enforces a one-to-one matching and is aware of the point density, while COV-CD and MMD-CD are sampling-insensitive.

## 5 EXPERIMENTS

Aware of both the sampling spectrums, we conduct experiments to further evaluate the performance of point cloud GANs under various evaluation metrics. We show that the point cloud GANs using sampling-insensitive discriminators may produce point clustering artifacts, while sampling-oversensitive discriminators fail to supervise point cloud GAN training at all. We further devise a diagnostic "no-generator" experiment that factors out the generators to better illustrate our discoveries on discriminators. More interesting, we find that the simple PointNet-Mix paired of any generator, even with the most naive fully-connected one, achieves the state-of-the-art performance.

### 5.1 SETTING AND DATASETS

We provide a thorough comparison of all the discriminators investigated in Sec.3 combining with all the available generators in the published literature, including the FC generator proposed in r-GAN (Achlioptas et al., 2018), and graph convolutional generators used in TreeGAN (Shu et al., 2019). We also add a deformation-based generator into the comparison given its popularity for supervised point cloud reconstruction (Groueix et al., 2018; Yang et al., 2018).

We use two datasets to evaluate the GANs. One is a single-category dataset containing point clouds sampled from all 6,778 chair meshes in ShapeNet (Chang et al., 2015). The other is a multi-category dataset combining shapes from airplane, car, chair, rifle, sofa, table, vessel categories in ShapeNet. The multi-category dataset contains 34,313 shapes in total. We uniformly sample 2048 points from each shape to form the two datasets. We follow the 85%/5%/10% train/validation/test split in (Achlioptas et al., 2018) and use WGAN-gp (Gulrajani et al., 2017) for the GAN training, similar to previous works (Achlioptas et al., 2018; Valsesia et al., 2018; Shu et al., 2019).

### 5.2 EVALUATING POINTNET-BASED DISCRIMINATORS WITH VARIOUS GENERATORS

We report the performance for point cloud GANs that combine PointNet-Max/Min/Attention discriminators and FC/Deform/TreeGAN/Graph-CNN generators in Table 3. We observe that GANs using PointNet-Mix as the discriminator outperform the ones using PointNet-Max/Attention across all different generators. On sampling-aware/sensitive metrics (*i.e.* FPD-Mix, FGD, MMD-EMD, COV-EMD), PointNet-Mix is always significantly better than PointNet-Max and is better than PointNet-Attention mainly on FPD-Mix and FGD. Regarding geometry quality evaluated using sampling-insensitive

| Dataset | Generator | Pooling | FPD-Mix ↓ | FPD-Max↓ | FGD↓ | MMD-E↓ | MMD-C↓ | COV-E↑ | COV-C↑ |
|---|---|---|---|---|---|---|---|---|---|
| Chair | FC | Max | 1.571 | 0.211 | 7.030 | 0.1017 | 0.00164 | 23.56 | 72.75 |
| | FC | Mix | **0.184** | **0.209** | **2.124** | **0.0674** | 0.00196 | 73.64 | 74.96 |
| | FC | Attention | 0.635 | 0.672 | 4.971 | 0.1156 | **0.00160** | 68.92 | 70.54 |
| | Deform | Max | 0.913 | 0.268 | 5.602 | 0.0908 | 0.00201 | 68.5 | 72.61 |
| | Deform | Mix | 0.534 | 0.373 | 2.836 | 0.0695 | 0.00200 | **76.29** | 75.11 |
| | Deform | Attention | 0.696 | 0.755 | 2.987 | 0.1141 | **0.00160** | 68.77 | 69.36 |
| | TreeGAN | Max | 1.442 | 0.654 | 7.808 | 0.0962 | 0.00191 | 24,74 | 73.49 |
| | TreeGAN | Mix | 0.293 | 0.334 | 4.032 | 0.0704 | 0.00211 | 74.82 | **78.79** |
| | Graph-CNN | Max | 1.034 | 0.981 | 7.494 | 0.0812 | 0.00191 | 48.90 | 63.18 |
| | PointFlow | | 1.782 | 1.405 | 3.256 | 0.1335 | 0.00190 | 71.76 | 71.73 |
| | Oracle | | 0.088 | 0.093 | 0.814 | 0.0594 | 0.00165 | 79.56 | 79.69 |
| Multi-Cat | FC | Max | 1.553 | 0.354 | 6.981 | 0.0842 | 0.00153 | 35.16 | 64.16 |
| | FC | Mix | **0.255** | **0.285** | 2.550 | **0.0656** | 0.00184 | **73.5** | **72.16** |
| | FC | Attention | 0.414 | 0.453 | 4.234 | 0.1188 | **0.00134** | 72.19 | 69.63 |
| | Deform | Max | 1.072 | 0.633 | 3.845 | 0.0799 | 0.00179 | 62.5 | 64.5 |
| | Deform | Mix | 0.614 | 0.349 | **2.451** | 0.0670 | 0.00191 | 70.83 | 68.83 |
| | Deform | Attention | 0.616 | 0.720 | 2.531 | 0.1113 | 0.00141 | 72.04 | 69.60 |
| | TreeGAN | Max | 1.714 | 0.437 | 6.342 | 0.1093 | 0.00158 | 25.33 | 67.0 |
| | TreeGAN | Mix | 0.388 | 0.420 | 4.300 | 0.0699 | 0.00184 | 72.66 | 71.0 |
| | Oracle | | 0.131 | 0.177 | 0.673 | 0.06012 | 0.00124 | 77.14 | 77.57 |

Table 3: **Evaluating PointNet-based discriminators with different generators.** We observe that PointNet-Mix can significantly improve all sampling-aware/sensitive metric, including FPD-Mix, FGD, MMD-EMD, COV-EMD for all the generators. When paired with FC generator, PointNet-Mix achieves the best performance. We include PointFlow(Yang et al., 2019) and Oracle for a comparison.

metrics (*i.e.* FPD-Max, MMD-CD, COV-CD), PointNet-Mix is always significantly better than PointNet-Attention on FPD-Max and COV-CD while remaining on a par with PointNet-Max.

In Figure 2, we present the generated point clouds for all the experiments with color-coding for the local point density. We see that the generators trained using PointNet-Max usually suffer from non-uniform density, except for the deformation generator. On the chair class, points are usually clustering around the seat area while being sparse at the back. On the contrary, PointNet-Mix enforces a globally uniform point density and hence greatly improves the visual quality of the generated point clouds. PointNet-Attention is in the between.

With PointNet-Mix, we observe that the most naive FC generator works the best outperforming the previously state-of-the-state generators on almost of metrics. This showcases the importance of being sampling-aware but not sampling-insensitive/oversensitive as a discriminator. It also suggests that future works may focus on designing more powerful sampling-aware discriminators.

## 5.3 DIAGNOSING FAILURES FOR VARIOUS DISCRIMINATORS

Observing that the discriminator forms the bottleneck of the point cloud GAN, it is important to examine whether advanced networks, *e.g.* PointNet++ (Qi et al., 2017b), DGCNN (Wang et al., 2019), KPConv (Thomas et al., 2019), PointConv (Wu et al., 2019) can outperform PointNet-based discriminator.

Surprisingly, we fail to observe any of them ever to be successful to train a GAN in our extensive experiments, in which we perform a systematic search, including varying generator architecture, changing hyperparameters (e.g. the number of layers and parameters in discriminator, learning rate), whether use batch norm, etc. During GAN training, we observe the four advanced discriminator behave very similarly: they are very discriminative in the sense that the gaps between the real and fake scores remain significantly large, but the quality of the generated point clouds stays very bad. The contradictory behaviors seem unreasonable at the first glance: how can a discriminator be very discriminative but teach nothing to the generator? Thus, we design a diagnostic "no generator" experiment to examine the underlying reasons.

**No Generator Experiments.** During GAN training, the gradients from discriminator output scores back-propagate to the generated point clouds and then further back-propagate to the weights of the generator to update the generator. We design a diagnostic experiment that removes the generator and examines whether the discriminator gradients are informative enough for supervising point cloud GANs.

Concretely, we take $M$ real chairs as real data and we randomly initialize $M$ learnable point clouds $p \in \mathbb{R}^{N \times 3}$ with i.i.d Gaussian noises $\mathcal{N}(0, 0.1)$. For each training iteration, we sample $B$ point clouds from the real dataset as the real data and sample $B$ point clouds from the learnable point

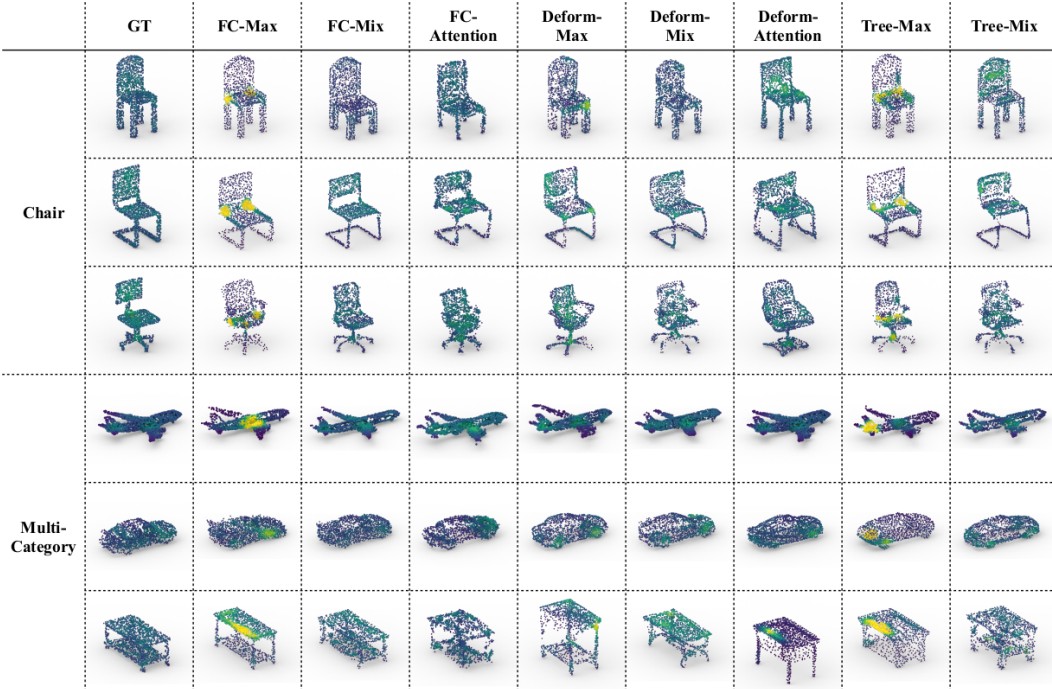

Figure 2: **Visualization of point clouds generated by different methods.** We show generated point clouds in few exemplar shapes for a fair and easier comparison regarding their sampling quality and geometry quality. We color-code the local point density that ranges from sparse (dark blue) to dense (light yellow).

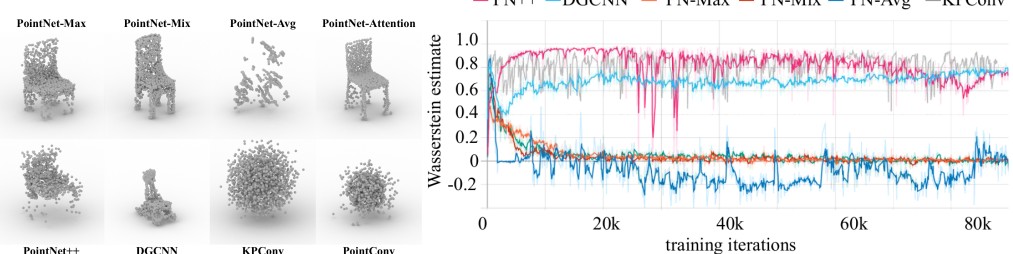

Figure 3: **The diagnostic "no generator" experiment results**. On the left, we show exemplar point clouds generated by different discriminators. On the right, we plot the training curves for the experiments. The $x$- and $y$-axis respectively represent training iterations and Wasserstein estimates.

| Discriminator | FPD-Mix ↓ | FPD-Max↓ | FGD↓ | MMD-E↓ | MMD-C↓ | COV-E↑ | COV-C↑ |
|---|---|---|---|---|---|---|---|
| PointNet-Max | 3.828 | 2.456 | 19.279 | 0.1039 | 0.00287 | 26 | **56** |
| PointNet-Avg | 19.640 | 19.604 | 39.793 | 0.1154 | 0.00642 | 32 | 12 |
| PointNet-Mix | 1.835 | 2.208 | 13.197 | **0.0898** | 0.00331 | 50 | 51 |
| PointNet-Attention | **1.350** | **1.642** | **12.359** | 0.1622 | **0.00271** | **52** | 48 |
| PointNet++ | 15.081 | 14.187 | 33.958 | 0.1155 | 0.00503 | 19 | 37 |
| DGCNN | 10.629 | 9.939 | 14.882 | 0.0932 | 0.00396 | 46 | 48 |
| KPConv | 41.15 | 49.96 | 57.32 | 0.2695 | 0.01135 | 6 | 2.5 |
| PointConv | 27.69 | 27.05 | 46.82 | 0.2832 | 0.00721 | 3 | 8 |

Table 4: **Quantitative evaluation for the diagnostic "no generator" experiments**. We see that PointNet-Max/Mix/Attention are successful while PointNet-Avg/PointNet++/DGCNN/KPConv/PointConv fail.

clouds as the fake data. Then, we conduct adversarial training where the discriminator gradients directly update the location of each points in the learnable point clouds. We adopt the same training protocol and losses as experiments in Sec. 5.2. The goal of this "no generator" experiment is to see whether the discriminator can modify the point clouds to approximate the real shapes, or in other words, we factorize the impact from generator architectures and only focus whether the gradients

from discriminators can guide shape generation. Here $M = 100$, $N = 2,048$, and $B = 32$. We train each experiment for 10,000 epoches until convergence.

**Results and Analysis.**    Table 4 summarizes the results and Figure 3 (left) shows exemplar generated point clouds. We clearly see that only PointNet-Max/Mix/Attention can successfully modify the learnable point clouds to get high quality shapes, while PointNet++/DGCNN/KPConv/PointConv/PointNet-Avg produce much worse results. Figure 3 (right) presents the training curves of the Wasserstein estimates, which essentially describe the gap between the real and fake scores. The large score gaps for PointNet++, DGCNN and KPConv throughout the training indicate the two models are very discriminative in telling apart the fake samples from the real data. However, their gradients simply don't help improve the learned point clouds. Note that both of them leverage relative point positions/edge information during their feature extraction, which leads to a huge amount of gradients flowing along the surface direction focusing on changing sampling patterns instead of supervising shape geometry. As a suggestion, for developing future discriminators, people may need to avoid using the relative position features. For PointNet-Avg, the failure is simply because the discriminating power is not sufficient, evident from the low Wasserstein estimates. Note that training such "no generator" experiment shares a similar flavor to the SGD sampling in introspective CNN(Jin et al., 2017).

## 6    CONCLUSION AND SUGGESTIONS FOR FUTURE DISCRIMINATOR DESIGN

In this work, we study the importance of sampling for 3D point cloud GAN design. We propose the sampling spectrum of discriminators and evaluation metrics that provide insights on the behaviors of point cloud GANs. We propose several empirical experiments for identifying the sampling sensitivity of a discriminator or an evaluation metric. Experiments indicate that, no matter what generator is employed, a sampling-insensitive discriminator, e.g. the commonly used PointNet-Max, will produce point cloud shapes with non-uniform density and clustering artifacts, while a sampling-oversensitive discriminator (*e.g.* PointNet++, DGCNN, PointConv, KPConv) will lead to disastrous failures when adapting them to training point cloud GANs. Interestingly, a simple PointNet-Mix baseline coupled with the most naive fully-connected generator achieves the best performance, indicating that the current bottleneck of point cloud GAN is on the discriminator side. For future discriminators, we suggest they should be more discriminative in shape and aware but not oversensitive to sampling patterns. For the sanity check of any novel discriminators, one can leverage our proposed "no generator" experiment.

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
