# OpenReview forum: "Rethinking Sampling in 3D Point Cloud Generative Adversarial Networks"
_ICLR.cc/2021/Conference — Reject_

### Official Review · AnonReviewer2 · 2020-10-27
**An interesting view in Point Cloud GAN. However, the work lacks theoretical background support and some technique details.**

**Rating:** 6
**Confidence:** 4

**Review:**

The paper conducts experiments to examine the effects of point sampling patterns in point cloud GANs. And, experimental results reveal the reason why current strong discriminators(PointNet++, etc) fail to train a reasonable point cloud generator. By proposing  a sampling spectrum, the authors introduce a middle point sampling-aware baseline discriminator--PointNet-Mix. The paper proposes a new improvement direction for Point Cloud GAN, which might have a strong impact in the community. The paper is written concisely and the illustrations are clear.

However, there are still some concerns that need to be addressed:

1. One of the major contributions claimed in the paper is the term “Sampling Spectrum”. However, the paper does not provide the exact definition of “Sampling Spectrum”. The paper mainly divides thes spectrum into three regimes: sampling-insensitive/-aware/oversensitive, which is oversimplified, and too broad. The authors should give more justification and background intuition on why the “Sampling Spectrum” should be in this form.

2. Another concern to this paper is the “no generator” experiments. In Figure 3, the authors show some visualizations of the point clouds generated by different discriminators. It is reasonable to generate some weird Chair shape from PointNet++, and DGCNN. However, the generated point clouds from KPConv and PointConv are quite surprising. To make the experiment results convincing, it is better that the authors should provide more details on the experiments of KPConv and PointConv, and provide more explanation on why KPConv and PointConv give almost random point clouds. Because the PointNet++, DGCNN, KPConv, and PointConv all use relative point positions/edge information according to the paper. There must be more reasons on why the results look very different.

3. More technique details on the “no generator”  experiments should be introduced. For example, what is the loss function used(both the “no generator” experiments and the rest GAN experiments)? Is it the improved WGAN?

---

> ### Author Response · Authors · 2020-11-21
> **Official Comment**
>
> Thanks a lot for your positive feedback. Here are our comments:
>
> * **the exact definition of sampling spectrum**
> As we discussed in the paper, sampling sensitivity theoretically should be defined as a measure of the rate at which the output of the discriminator changes with respect to the change in the input sampling pattern. However, there is no easy way to measure the change in the input sampling pattern. We have to discretize the change and categorize different sampling sensitivity regime.
> We take the two key abilities of a discriminator as landmarks for measuring their sampling sensitivity: 1) whether the discriminator can tell clustering artifacts, where a lot of points are distributed in a small region on the object surface with out-of-distribution high density 2) whether the discriminator can distinguish FPS and uniform sampling if the two set of point clouds share the same set of shapes.
> Our objective of studying sampling effects is to help build a better point cloud GAN. The reason we care about clustering artifacts is that this artifact can significantly affect the visual quality of generated point clouds. We believe the ability to tell this is the minimum requirement for being a sampling-aware discriminator. For telling apart FPS and uniform sampling, the choice of FPS and uniform sampling are not a must but the ability to distinguish these two will be an overkill for a discriminator, which means the discriminator will score high only if the generated point clouds further mimic the exact sampling pattern even if the generated shape is already perfect. For GAN, such a discriminator will make the generator hard to converge. We actually find in ''no generator" experiments that the gradients of such generator are chaotic. Based on this observation, we update the paper to incorporate an empirical sampling sensitivity measure, please take a look at Sec. 3.2.
>
> * **no generator experiment**
> The loss we used is WGAN-gp and it shares the settings with other GAN experiments. In sec. 5.3, we updated our paper to include more details regarding the experiment setting.
>
> * **KPConv and PointConv in no generator experiments**
> KPConv and PointConv all leverage a learnable kernel for point convolution, which encourages spatial-anisotropic processing of edge information. In contrast, PointNet++ and DGCNN can only learn a spatial-isotropic kernel to process all edge information, where their kernel is not edge-dependent. So, KPConv and PointConv are more powerful to extract local/edge information and hence they are even more sensitive to sampling patterns than PointNet++ and DGCNN, as shown in Table 1.  This supreme sampling sensitivity allows them to beat generators by only looking at local edge information without even looking at the global shape information and hence significantly make their gradients biased toward fixing sampling. On the contrary, under normal GAN training, PointNet++ and DGCNN are not powerful enough to completely ignore the global shape.
> When KPConv and PointConv only focus on fixing the local sampling pattern, think about the difference between uniform sampling and FPS,  it would be extremely hard for them to exactly clone one sampling pattern locally. That's why KPConv and PointConv are moving points back and forth without doing anything meaningful to improve the shape generation.

---

### Official Review · AnonReviewer1 · 2020-10-28
**Insightful take on the generative capabilities of various point cloud GANs.**

**Rating:** 7
**Confidence:** 5

**Review:**

This paper considers the task of 3d shape generation. More precisely, it embraces the point cloud GAN strategy for generation and offers an extensive comparative analysis of the existing architectures of point cloud generators and discriminators with a focus on the variations of the latter component.

A series of synthetic experiments considering several point cloud sampling schemes is proposed and used to define a spectrum of sensitivity to sampling artifacts (various deviations from uniform sampling) for a range of considered discriminators. The same is performed with the existing metrics evaluating performance for point cloud generation.

Presented results suggest that existing PointNet based discriminators with max-pooling point feature aggregation mechanism are insensitive to severe artifacts in point density, which leads to non-uniform samples from the generators. On the other hand, existing improvements to PointNet, which perform better in discriminative tasks are over-sensitive to sampling and manage to capture even the smallest deviations from the uniform sampling which leads to poor training signals (in terms of gradients) from those discriminators. As alternatives for both extremes the authors propose to use «sampling-aware» PointNet based discriminators, but either use a concatenation of the max-pooled and average-pooled point features, or self-attention mechanism as an aggregation function.

In the experimental section the authors compare numerous combinations of considered discriminators with several existing generators to confirm that the proposed «sampling-aware» discriminators yield superior performance across all the generators.

Pros:
1) The paper is well-written and is easy to follow.
2) Overall rigorousness of the experiments in the paper is impressive, and the conclusions drawn are logical and insightful.
3) Additional inputs on the sampling sensitivity spectrum for evaluation metrics are also valuable.

Cons:
1) The main drawback of the paper is the self-contained nature of the presented material. In its current form, it misses any validation by comparison to any external generative results. Thus, it is not clear, where the obtained improvements for point cloud GANs put them on the performance quality list, considering all the generative models for point clouds.

Overall, this is an impressive work, which I think may be accepted even as it is. However, its value can be extended outside point cloud GAN community to general 3d shape generation task audience with a proper comparison of the best results with current state of the art. There are at least two recent works based on likelihood training [1, 2], which can be considered for comparison, or at least should be mentioned. Given that my score is only moderate.

#################################################

Additional comments:

Section 1, paragraph 2:
Improper citation caused artifacts in the text.

Section 5.1:
Both CD and EMD (also possibly FPD) are sensitive to the scale of the input point clouds, thus it is important to indicate the scale of the input point clouds (how they were normalized). For example for now, I am not sure that it is possible to directly compare your results from Table 3 with the results of [1, 2], since your values of MMD-CD/EMD and COV-CD/EMD significantly differ from the values reported there. This might be due to the different input data scales.

Section 5.2:
In my experience, l-GANs from [3] gave very unstable performance with respect to even small variations in the model architectures (e.g. a change in the dimensionality of the latent space). To obtain reasonable performance I needed to tune hyperparameters related to training objective and optimization separately for different configurations and could not use a single set of hyperparameters for all the configurations.
In your case, you alter different network parts completely. Could you verify somehow that the relative performance of the considered configurations can not be attributed to the choice of the hyperparameters, which are best for one configuration but not optimal for others? In other worlds, have you tried to tune the hyperparameters separately for different configurations?

I appreciated the PointNet-Max-2048 experiment in the supplementary materials, since I had the same question and found the answer there.

Section 5.3, No generator experiments:
«…to supervised the generator training.» -> to supervise

Figure 2 is reported to contain «random selected» (-> randomly selected) samples. I do not understand how they can be randomly selected and at the same time all present approximately the same object in each row. I do not think you can claim random selection rather than cherry picking in that case, which is not a problem, since you wanted to demonstrate the differences in generations of approximately the same object.

Table 4, caption:
«Min» -> Mix

[1] Yang, G., Huang, X., Hao, Z., Liu, M.Y., Belongie, S., Hariharan, B.: PointFlow: 3D point cloud generation with continuous normalizing flows. In ICCV’19.

[2] Klokov, R., Boyer, E., Verbeek, J.: Discrete Point Flow Networks for Efficient Point Cloud Generation. In ECCV’20.

[3] Achlioptas, P., Diamanti, O., Mitliagkas, I., Guibas, L.: Learning representations and generative models for 3D point clouds. In ICML’18.

___

After reading other reviews, authors' comments, and checking the revised manuscript I decided to slightly improve my rating for two reasons. Firstly, my concerns were answered during discussions, secondly, I do not agree that the concerns raised by other reviewers could justify a rejection. I believe this is an exemplary empirical study presenting novel information and insights about sampling sensitivity of point cloud encoders and point cloud evaluation metrics.

---

> ### Author Response · Authors · 2020-11-21
> **Official Comments**
>
> Thanks for your positive feedback. Here are our comments:
>
> * **comparison with PointFlow**
> We incorporate the results from PointFlow into Table 3 for a comparison. We are essentially using the same training data with same normalization protocol. Note that the values from Table 1 in PointFlow are not of the original scale. They wrote in the table caption, "MMD-CD scores are multiplied by $10^3$; MMD-EMD scores are multiplied by $10^2$". For MMD-CD/EMD, COV-CD/EMD, we directly take their values in the original scale and put them into our Table 3. For Frechet distance based method, we evaluate their generated shapes under the same protocol as us. Our results show that PointNet-Mix outperforms PointFlow on all metrics. Actually, PointFlow takes Gaussian noise point clouds as input and use CNF to move those points, which leads to a lot of noise sampling patterns in the final results. It is easily to understand why our generated point cloud supervised by sampling-aware PointNet-Mix can beat them.
> * **Hyperparameters**
> In this work, we don't consider the l-GAN which performs discrimination in the latent space. Instead, we go for the r-GAN and improve it by changing the discriminators. Our extensive experiments show that r-GAN is not sensitive to hyperparameters and the discriminator actually plays the most important role. When we use PointNet++ and DGCNN as discriminators, we actually perform a systematic hyperparameter search, including the number of layers and parameters in discriminator, different generator architecture, learning rate, whether use BN, etc. But, we couldn't find any settings under which PointNet++ and DGCNN are functional, the same applies to KPConv and PointConv. That's exactly the reason why we want to understand the failure mechanism of these advanced discriminators, from which we find the sampling sensitivity issue. So, we believe we already outperform PointFlow by a big margin and make a scientific contribution on examining different discriminators. Hyperparameter tuning won't change these two contributions.
> * **random selected vs. cherry pick**
> Thanks for pointing it out. We acknowledge that this is a bit confusing. We already make it clear in the caption that we on purpose to select exemplar shapes for comparison and remove the confusing claim on "randomly selected".
>
> Thanks for the careful reading. We fix all the other typos and artifacts you mentioned. We believe this work is with both performance achievements and scientific values and hence is worthy of publication in ICLR. Given the current reviews, it would be crucial for you to give us stronger support. We would like to further address any of your concerns.

---

> > ### Comment · AnonReviewer1 · 2020-11-21
> > **Comparison to oracle metric values is missing and strange.**
> >
> > Thank you for your comments, I appreciate the change in the caption of Figure 2 and the explanation about the hyperparameters. I think the fact that you "... couldn't find any settings under which PointNet++ and DGCNN are functional, the same applies to KPConv and PointConv" in terms of all the various network and optimization hyperparameters deserves to be expressed somewhere in the paper.
> >
> > ***
> >
> > Thank you for pointing out the fact that PointFlow used scaling coefficients for MMD-CD/EMD, but I was aware of that and did not mean that difference when I wrote that "... your values of MMD-CD/EMD and COV-CD/EMD significantly differ from the values reported there." Let me elaborate on my concerns here. You stated in the reply, that you use the same normalization of the input data as the PointFlow paper. This implies that your values should be directly comparable to the PointFlow values. Let me compare the metric values for your FC generator MIX aggregation model with the values for PointFlow and oracle model.
> >
> > |Model|MMD-CD|MMD-EMD|COV-CD|COV-EMD|
> > |---|---|---|---|---|
> > |yours|0.00196|0.0674|75.0|73.6|
> > |PointFlow|0.00242|0.0787|46.8|47.0|
> > |oracle|0.00192|0.0738|57.3|55.4|
> >
> > For 3 out of 4 considered metrics your results exceed the oracle performance. For COV metrics the margin is especially large. This implies, that your model is capable of generating samples that are generally closer to the samples from the test set than the samples from the training set. Explanations for exceeding MMDs were proposed in [1, 2], but for COV it is unique behavior. There is either a mistake in the COV calculations or some other issue that should be investigated and explained.
> >
> > ***
> >
> > "Actually, PointFlow takes Gaussian noise point clouds as input and use CNF to move those points, which leads to a lot of noise sampling patterns in the final results. It is easily to understand why our generated point cloud supervised by sampling-aware PointNet-Mix can beat them.":
> >
> > A description of PointFlow approach does not explain why your method works better. Flow-based approaches use completely different generator architectures and training objectives and it is far from clear or easy to understand why one is experimentally preferable over the other (especially given that l-GANs and PC-GAN empirically perform worse than flow-based likelihood approaches).

---

> > > ### Author Response · Authors · 2020-11-23
> > > **Adding Oracle Scores**
> > >
> > > We updated Section 5.3 to include that we have already searched for architecture and hyperparameters. Let's know if this is clear right now.
> > >
> > > Regarding the comparison with PointFlow and oracle, we find that our training/test split is slightly different from the one used by PointFlow. We used the split file provided by the author of [3] whereas PointFlow has its own split. For example, on the chair category, our split is train 5761 vs test 679, whereas PointFlow is train 4612 vs. test662.
> > >
> > > On our test data, we compute the oracle score and updated in Table 3. It can be seen that we don't exceed the oracle on the COV-CD and COV-EMD metrics.
> > >
> > > In comparison with PointFlow, we have more training data but keep a similar amount of test data. Due to the long training time required by PointFlow, we can't finish training PointFlow on our training split by the end of rebuttal period. We then use PointFlow's checkpoint and test its performance on our test split. Please look at Table 3. We acknowledge this is not a fair comparison. However, given that PointFlow is way worse than our method on FPD-Max and FPD-Mix, we believe this margin is significant enough to say that we are highly likely to be better than PointFlow on Frechet scores. We will update the results for PointFlow in the camera-ready version if this paper can be accepted.

---

### Official Review · AnonReviewer3 · 2020-10-29
**review for ICLR2021 submission "rethinking sampling in 3D point cloud generative adversarial networks"**

**Rating:** 4
**Confidence:** 4

**Review:**

This paper studies an interesting problem, i.e., point sampling, in 3D point cloud generation through GAN. It shows that the currently used discriminator in this framework is sampling insensitive, thus the learned generator is prone to sampling artifacts. On the contrary, the sampling sensitive point-cloud CNNs are not suitable for acting as a discriminator according to the authors’ experiments that all of them fail to generate reasonable point clouds. Observing this, the authors proposed sampling-aware discriminator and achieved better results. Especially, along with the proposed sampling-related metrics, the proposed method shows advantages over existing methods.

Strength:
1. The finding that sampling plays an important role in GAN based point cloud generation is helpful to the community.
2. The proposed method by simply concatenating an average pooled feature to the max-pooled PointNet feature is shown to be effective in the proposed metrics.


Weakness:
Overall, there are many unclear issues need further explanation. The rationality of the proposed method is also unconvincing.
1. Intuitively, as the sampling insensitive discriminator leads to generated point clouds unaware of the sampling artifacts, the advanced sampling sensitive discriminator could address this problem since it focuses more on this aspect. However, the results did not support this guess and the paper does not give strong reason/analysis to explain such failures.
2. The proposed sampling aware discriminator is a concatenation of max pooling and avg pooling. Considering PointNet-MIX and PointNet-MAX only differs by an avg, does it mean that avg is sampling-aware? then, why not directly using avg? Unfortunately, the results shown in the paper (e.g., table 4) demonstrate that using avg also does not perform well. Therefore, it is quite hard to understand the underlying rationality for the success of PointNet-MIX.
3. To show the proposed mix strategy is general enough, it is required to show with other baselines. For example, how about using the mix strategy in aggregating weighted feature in PN-attention? Does it still work?
4. PN-attention improves PN by incorporating communication between points, however, why other networks such as PN++, PointConv that also uses points interactions can not improve on PN in GAN? This is worthy to analysis and give a reasonable explanation in this paper.
5. About the proposed sampling-related metrics, the core is F distance based on various feature extractors. Since F distance is based on features, while the used features are depended on extractors, so this observation in P5 “the Frechét distance metrics share the same sampling sensitivity of their corresponding discriminators.” is obvious. In other words, it actually heavily relies on the used extractor, i.e., with a sampling insensitive extractor, the metric is tend to be sampling insensitive. Therefore, such metrics may be meaningless to reflect the property of different methods/generators.

Other comments:
It is hard to understand what is “learnable point clouds” in No Generator Experiments. More details about the setup and network architectures about this experiment is helpful for reader to understand.

---

> ### Author Response · Authors · 2020-11-21
> **Official Comment**
>
> Thanks for your feedback. Here are our comments:
>
> * **Why advanced sampling sensitive discriminators fail?**
> We believe there is probably a misunderstanding that we are solely talking about fixing sampling errors for the advanced sampling-sensitive discriminator. The most important fact in this work is that sampling-sensitive discriminators fail to generate anything meaningful regarding both geometry and sampling. We write down this fact in the first paragraph of Section 5.3. (also see Fig 3 for visualization). The question of why these advanced discriminators can't fix clustering artifacts is not meaningful, since they even can't generate an okay shape. We will explain why this happens along with no generator experiments as below.
> * **No generator experiment**
> We investigate why advanced discriminators fail in Section 5.3 by using no generator experiment and provide a detailed analysis in the last paragraph of this section. We are sorry that the experimental setting of no generator was not clear. We already updated our paper and hope it is perceivable now. Please read the text again and let us know if it is still unclear. We here further provide two explanations that may help you understand better.
>   1)**Intuitive explanation**: you can think the discriminator is a teacher, the generator is a student, and the generated shapes are the homework submitted by the student. Our argument is that to improve the student's learning, the teacher needs to point out the errors in the homework (discriminator gives the gradients to the generated shapes) and then the student can learn (gradients backpropagate to update generator's weight). We can roughly make an analogy by saying that geometry is the correctness of solutions in the homework and sampling is the handwriting quality of the homework. If the teacher cares too much about the quality of handwriting (a discriminator is sampling over-sensitive), the teacher may always say your handwriting is bad and I don't even want to check whether your solution is correct. Why the teacher can't provide guidance on both problem solutions and handwriting quality?  Because the training objective of the teacher (the discriminator) is just to tell apart the ground truth solution and the homework submitted by the student. If the teacher can tell from the handwriting that the homework is not printed on the paper, then the teacher doesn't need to check the answer at all (in this case the discriminator will only give feedback regarding the sampling patterns without caring about the quality of the generated shapes). Again, you will never see a student write like printed text, this is exactly the reason why the subtle sampling error can't be fixed by our discriminators (think about how hard it is to mimic furthest point sampling patterns in generated point cloud). Without fixing the subtle sampling difference, we believe the discriminator will not proceed to supervised geometry generation. We design the no generator experiment to check if we get rid of the student but only submit the homework to the teacher, will the teacher correct the errors in the submitted homework? We believe if the teacher can't correct the errors in the homework regarding problem solutions, then no student, regardless of what kind of this student is, can learn from the teacher.
>   2)**technical version**: the role of no generator is to factorize the impact of generators from GAN training. We find that any discriminators that are sampling-oversensitive will fail in the no generator experiments and therefore their gradients are focusing on correcting sampling patterns, which will never be fixed, and ignoring geometry difference. Why does this happen? As a discriminator, the training objective is to tell generated shapes from real shapes.  For sampling-sensitive discriminators, e.g. PointNet++, DGCNN, and KPConv, think about how these advanced discriminators are built, which decides how they make the real/fake decision. They extract edge features in a hierarchical way with small receptive at early stages. We argue that those edge features are sensitive to sampling patterns, which enables them to tell the real/fake locally at the very first few layers. However, perceiving geometry is a problem with non-local scope, only if the features are not sensitive to sampling, the decision will be made according to the global geometry. That's why the gradients of these discriminators prefer to fix sampling patterns because their discrimination is done locally from the first few layers and there is no need for them to perceive global geometry, which means no way to improve geometry. This is exactly what we want to show in the no generator experiment, where any discriminator that can tell FPS from uniform sampling will fail and then fail in point cloud GAN.

---

> > ### Author Response · Authors · 2020-11-21
> > **Cont'd: Official Comment**
> >
> > * **Avg pooling**
> > Average pooling is sampling aware. As we explained in Section 3.2, the average pooling feature computes the center of the mass of the points in feature space and is hence aware of certain global non-uniform density distributions, e.g. if a lot of points are located densely around one point, then this can change the average feature significantly. However, sampling-aware is not enough. To be a successful discriminator, it first needs to be salient about geometry and then it will generate more uniform distributed point clouds if it is further sampling-aware (but not sampling-oversensitive). The reason why PointNet-Avg fails is that it is not geometry-salient. Think about the average operation: the locations of the center of mass, which is the average center of point clouds, can be the same for two different shapes; similarly, the averaged feature can be the same for two different shapes. We believe this shows that PointNet-Avg is too weak as a good generator to supervise geometry generation. This is proved in no generator experiment that the gradients of PointNet-Avg are not helping shape generation. Also, see the right figure in Fig 3, PointNet-Avg is the only discriminator that has a negative Wasserstein distance, which is (the score of real - the score of fake). This means PointNet-Avg is too weak and easy to be fooled (in another words, average pooling is vulnerable to adversarial attacks). We briefly explained this in the second last sentence in Section 5.3.
> > To explain the success of  PointNet-Mix, it basically takes the best of both worlds from a geometry-salient discriminator, PointNet-Max, and a sampling-aware discriminator, PointNet-Avg, by concatenating their features.
> >
> > * **Mix pooling and PointNet-Attention**
> > Through our experiment, we found that mix pooling doesn't affect PointNet-Attention's performance (neither improves nor hurts). The reason is that the average pooling will be beneficial only if the discriminator is currently sampling-unaware, e.g. PointNet-Max, in which case the average feature will help to uniformly distribute the generated points and hence improve EMD-based metrics a lot. As we show in the Table 1, PointNet-Attention itself is already sampling-aware. So, adding average feature to it will not improve it. In nature, PointNet-Attention performs a learned weighted average, this is why it is not as robust as max-pooling and its performance is not the best in Table 3.
> >
> > * **PointNet-Attention vs. PointNet++ and DGCNN**
> > Although PointNet-Attention and PointNet++ both encourage point-to-point communication, they do it in a completely different way. Attention mechanism learns a global weighted matrix to fuse point features but it doesn't use local features or edge features and doesn't perform hierarchical feature extraction. As I explained in the comment above, edge features are very sensitive to sampling patterns, and, due to the hierarchical feature extraction, PointNet++/DGCNN/KPConv can make an early real/fake decision leveraging edge features while PointNet-Attention can't. This is the reason why PointNet-Attention can pass no generator test while all the advanced discriminators fail.
> >
> > * **sampling-related metrics**
> > We fully agree with your opinions on how the feature extractor affects the role of a metric. However, in this work, we are not only caring about the sampling quality but also geometry quality. When a metric is sampling insensitive, e.g. FPD-Max, it is not necessarily meaningless. On the contrary, it means this metric only focuses on shape quality and should be considered if one doesn't care about sampling. Note that FPD-Max is not proposed by us but in ICCV 2019 work, 3D Point Cloud Generative Adversarial Network Based on Tree Structured Graph Convolutions (it is called simply FPD in this work). In addition to FPD-Max, we propose FPD-Mix and FGD. Our contributions are to provide more choices to accommodate certain needs to enforce sampling pattern. If you don't want to have clustering artifacts, please use FPD-Mix. If you even care about the difference between FPS and uniform sampling, then you should use FGD (extract by DGCNN).

---

### Official Review · AnonReviewer4 · 2020-11-02
**novel studies on how point clouds’ sampling strategy affects generation quality and evaluation, needs work on the “sampling spectrum” framework to organize the results**

**Rating:** 6
**Confidence:** 5

**Review:**

Summary of the paper:
This studies how different discriminator architectures are sensitive to sampling strategy of point clouds.
The paper proposes three benchmarks that measures how sensitive the architecture to the sampling strategy of point clouds, which provides insights to future research about how to choose architectures for both discriminator and for feature extraction.
Empirically, the authors find out that using a discriminator architecture that’s just reasonably sensitive to sampling could create better generative results, regardless of the generator.
Finally, the paper also provides a novel experiment setting (i.e. “no-generator” experiment) that could compare gradient quality of discriminators.


Strength:
This work is the first one (that I’m aware of) that studies how the point sampling scheme affects the results and evaluation of the generation. The paper also tried to verify such results with a wide ranges of architectures and the results seemed rather self-consistent.

Weakness:
1. Discreteness of the “sampling spectrum”. While the idea to study how sensitive the network is to the sampling schema is certainly good, but the way to characterize the spectrum in this paper is too empirical and discrete. This might not be too useful when the sampling method of interested in certain application doesn’t fall into the category the author mention. It would be nice if the author could provide a way to quantify the difference between different sampling methods (e.g. expectation of EMD on the same surface area), and verify how different architecture falls into such continuous spectrum.
2. Disentangling the effects of geometries v.s. the sampling. I agree with the authors that the point clouds’ quality can be thoughts as outputs of the geometries and the sampling, and those two factors goes hand-in-hand. But this paper seemed to emphasize solely on the architecture’s capacity to detach different sampling techniques, and tries to correlate this factor with the generation quality. My concern here is that could the network’s capacity toward geometries a confounding factors in the experiment results? For example, could it be the case that the network that’s over-sensitive to the sampling patterns actually fails to tell apart geometric difference, which leads to bad generative results. In that case, could it be the case that the network’s ability to discriminate geometries differences the main factor, regardless of whether the network is over-sensitive or under-sensitive to the sampling? Following this line of thoughts, if the authors’ claim holds, then there might be some fundamental difficulty for the network to be both capable of telling apart geometries and being over/under-sensitive to the sampling pattern, which doesn’t seem to be established in the paper concretely (with theory or empirical results). Maybe the authors could point us to evidence that show that the networks used in the paper all have about the same capacity in telling apart geometries, which is an alternative to address this concern.


Justifications:
The paper is the first in my knowledge that studies how sampling strategy affects point cloud generation and evaluation. In this case, the paper has its value and novelty in the literature. But the results and insights are organized under the framework of “sampling spectrum”, which might still need improvement with more theories or results. WIth that, I will vote for borderline + for now and will be happy to hear from the authors.

---

> ### Author Response · Authors · 2020-11-21
> **Official Comment**
>
> Thanks a lot for your positive feedback. Here are our comments:
> * **Discreteness of the "sampling spectrum"**
> We acknowledge that this paper is an empirical study of the effect of sampling sensitivity of a discriminator on training Point Cloud GAN. We assess the sampling sensitivity of a discriminator mainly regarding their ability to tell apart three exemplar sampling patterns, e.g. uniform sampling with clustering artifacts, uniform sampling, FPS. Although simple and incomplete, the discriminative power regarding these three specific patterns is crucial and sufficient to predict the behavior of the discriminator given that we are only interested in improving point cloud GAN other than building a theory for sampling pattern analysis. We here explain why these three sampling patterns are particularly important for us:
>   1)  Clustering artifacts, where a lot of points are distributed in a small region on the object surface with out-of-distribution high density, can significantly affect the visual quality of generated point clouds. We believe the ability to tell this is the minimum requirement for being a sampling-aware discriminator.
>   2) For telling apart FPS and uniform sampling, the choice of FPS and uniform sampling are not a must but the ability to distinguish these two will be an overkill for a discriminator, which means the discriminator will score high only if the generated point clouds further mimic the exact sampling pattern even if the generated shape is already perfect.
>
> In our paper, we have shown that if a discriminator can tell the clustering artifacts, it can enforce a uniformly distributed point cloud generation, see Figure. 2; if a discriminator can tell FPS from uniform, it will fail in the no generator experiment and hence fail to be a functioning discriminator regardless of the generator. So, we found this discrete depiction is almost enough for studying point cloud GAN.  We updated the section 3.1 by incorporating a continuous measure, called empirical sampling sensitivity (ESS) to avoid being completely discrete, which now forms a truly continuous spectrum. Please check and give us your feedback.
>   In this work, beyond studying point cloud GAN, we were hoping to build a theory to measure the difference between sampling patterns. However, this is way more tricky than it seems to be. Regarding your suggestion on "quantifying the difference between different sampling methods (e.g. expectation of EMD on the same surface area)", we have shown in Table 2 that the EMD between different sampling patterns is not informative for telling apart patterns: For a set of uniformly sampled training point clouds, resampling again will generate another set of point clouds with an even larger MMD under both EMD and Chamfer metric, comparing to point clouds sampled by FPS (the collection of shapes are fixed). Actually, EMD or Chamfer only consider the first-order spatial distribution of a point cloud, second-order pairwise point distance and higher-order statistics are also important for characterizing a sampling pattern. Since the sampling pattern should be modeled as a joint position distribution of all points, which is very high-dimensional, it is very difficult to use a low-dimensional way to fully characterize it. We have searched around and failed to find any literature that builds a systematic analysis regarding point cloud sampling pattern distribution and measuring the difference between sampling patterns. We believe this is beyond the scope of understanding sampling sensitivity for point cloud GAN and will leave this topic for future works.

---

> > ### Author Response · Authors · 2020-11-21
> > **Official Comments**
> >
> > * **Disentangling the effects of geometries v.s. the sampling**
> > Point cloud is an entangled representation of geometry and sampling. Although we emphasize the discriminative power against different sampling patterns a lot, geometry is the one that goes first. At the very beginning of this project, we observe that for PointNet++, DGCNN, KPConv, and so on as classifiers, they all show a better performance than PointNet on geometry classification on ModelNet40 benchmark (the sampling pattern is fixed as FPS). Our motivation is to leverage their better geometry discriminative power to improve current point cloud GAN. However, our experiments show that they fail significantly as a discriminator compared to PointNet. So,  the truth is that PointNet++, DGCNN, KPConv can better discriminate on both geometry (shown by ModelNet40 object classification benchmark) and sampling (shown by us in Table 1) compared to PointNet. Hence, an ideal discriminator is that it can discriminate geometry while not being sampling over-sensitive. We found such discriminator exists, e.g. PointNet-Max, PointNet-Mix. The pity is that all existing advanced point cloud networks beyond PointNet are sampling over-sensitive. We explain the reason in "Results and Analysis" under Section 5.3. We want to provide a detailed explanation of why they fail here again: As a discriminator, the objective is to tell generated shapes from real shapes. Think about how these advanced discriminators are built, which decides how they make the real/fake decision. They extract edge features in a hierarchical way with small receptive at early stages. We argue that those edge features are sensitive to sampling patterns, which enables them to tell the real/fake locally at the very first few layers. However, perceiving geometry is a problem with non-local scope, only if the features are not sensitive to sampling, the decision will be made according to the global geometry. That's why the gradients of these discriminators prefer to fix sampling patterns because their discrimination is done locally from first few layers and there is no need for them to perceive global geometry, which means no way to improve geometry. This is exactly what we want to show in the no generator experiment, where any discriminator that can tell FPS from uniform sampling will fail and then fail in point cloud GAN.
> >
> > Our study gives guidance for building future discriminators that should ideally outperform PointNet on geometry classification while avoiding using sampling-sensitive features. We sincerely hope our argument can convince you that we perform a pioneering study in this field even if we lack a theory for sampling pattern distribution, which itself deserves papers to specifically investigate. We believe our work is worthy of publication in ICLR and hope our work can attract more attention to this field. Given the current reviews, it would be crucial for you to give us stronger support. We would like to further address any of your concern.

---

### Official Review · AnonReviewer5 · 2020-11-06
**Why not convolutional generator?**

**Rating:** 5
**Confidence:** 3

**Review:**

This paper experimentally examine a number of generator and discriminator network choices for point cloud GAN. It is shown that the best generator choice would be a PointNet with a mixture between max-pooling and average pooling, and that an attention-based PointNet framework performs the best in terms of discriminators. Multiple metrics were evaluated and a comprehensive experiment was done on the metrics capabiliities w.r.t. the sampling procedure.

Positive:

-- Comprehensive experiments, multiple metrics, network structures tested on both the generator and the discriminator side.

Negative:

-- I don't get why the authors would always stick to existing generators. It has been shown that convolutional approaches such as KPConv and PointConv perform much better than PointNet in discriminative settings. Hence, it also makes sense that they could perform well in a generative setting. The tested graphCNN generator is not necessarily as powerful as KPConv/PointConv. From an intuitive point of view, convolutional architectures should be helpful in the generative models as well. Besides, using a convolutional discriminator may not be able to generate gradients good enough for a non-convolutional generator. But if the generator and discriminator match in terms of architecture presumably the performance could be better.

-- The other concern I had is w.r.t. detailed settings of the "no generator" experiment in Table 4. Is WGAN-GP-type cost-term and penalty used in this experiment? It's well-known that discriminators that are very confident would not give proper gradient to generative models (e.g. from the WGAN papers, I also remember reading somewhere this paper "A good GAN requires a bad discriminator" although somehow I couldn't find this reference anymore), unless they are heavily constrained e.g. by WGAN-GP kind terms. Hence this detail can be important in deciding whether the conclusions from the paper would be credible.

The main reason I want to nitpick on these seemingly small items is that the conclusion of the paper might change significantly from those details. The paper tends to get to a strong conclusion that one needs a bad discriminator for a good GAN, although that's generally a reasonable assumption, its conclusions on inability to generate anything with a KPConv/PointConv discriminator could have far-fetching implications and hence I just want to apply a bit of additional scrutiny here.

Minor: For completeness, it's worth writing down what is the input to the point cloud GAN. This is very unclear in the current paper. I assume that the input is a random point cloud as in (Achlioptas et al. 2018), but it's worth stating that clearly early on, e.g. Sec. 2.1.

---

> ### Author Response · Authors · 2020-11-20
> **Clarification to generator input and why not using convolutional generator**
>
> * **"what is the input to the point cloud GAN"** \
> The input to our point cloud GAN: our generators take input a random latent vector $z \sim \mathcal{N}(0, I)$ and output a point cloud $p \in \mathbb{R}^{N\times 3}$. This is the standard setting used in normal GAN, e.g. image GAN and all existing 3D point cloud GANs. As far as we know, Achlioptas et al. 2018 also used this setting. I think a flow-based model, e.g. PointFlow, may take random point clouds as input. But this is a significantly different problem from what we are investigating in this paper. We updated Section 2.1 to make this point clear.
>
> * **Detailed settings of the "no generator" experiment**\
> For your second question, we do have the gradient penalty for "no generator experiment". Except for no generator, this experiment shares all the other settings with our main experiments in Sec. 5.2 (we have WGAN-gp loss). We updated our "no generator" section to make this more clear. So, we believe our experimental results are solid.
>
> * **"why not convolutional generator"** \
> Regarding your first question: you suggest us to use KPConv and PointConv in our generator, because "using a convolutional discriminator may not be able to generate gradients good enough for a non-convolutional generator". We highly appreciate your idea regarding how to improve generators. But the issue you raised up, incompatibility between discriminators' gradients and generators' architecture, is a secondary requirement for training a good GAN. Before that, the gradients from the discriminator should be correct and informative for shape generation. As we argued in the "no generator" experiment, to make a generator successful, the gradients from the discriminator should first update the generated point cloud correctly, whose gradients will backpropagate to update the generator's weight. Basically, if a discriminator passes no generator experiment, it then needs to be compatible with the generator to train a good GAN; if a discriminator fails in no generator experiment, regardless of the generator architecture, it will fail in training GAN. This conclusion is always true throughout all of our experiments in section 5.3 and, hopefully, this can convince you that changing the generator will not save a discriminator who fails in no generator experiments. Given that KPConv and PointConv both fail in the no generator experiment, we believe their gradients to point clouds are distracted by fixing the sampling pattern and fail to improve the shapes due to their sampling sensitivity, and hence no generator will be able to save them.
>
> Hopefully, our rebuttal addresses your concern. This paper is with a lot of technical details and its logic may be a bit confounding at the first sight. But we believe it carries significant scientific values that can benefit point cloud GAN research. We sincerely hope you can support us more and help us to get this paper published. We would like to address any further concerns you have.

---

### Decision · Program_Chairs · 2021-01-07
**Final Decision**

**Decision:**

Reject

**Comment:**

The paper provides empirical evidence that the sampling strategy used in point cloud GANs can drastically impact the generation quality of the network. Specifically, the authors show that discriminators that are not sensitive to sampling have clustering artifact errors, while those that are sensitive to sampling do not produce reasonable looking point clouds. They also provide a simple way (i.e. including AVG feature pooling) to improve generation quality for insensitive discriminator GAN setups. The reviewers agree that this is an interesting insight into the problem and this insight can help the community.

Based on the reviewers' comments and subsequent discussions, it becomes clear that the paper would be stronger and more compelling if the underlying hypothesis (i.e. the idea of sampling spectrum) is more rigorously defined (e.g. ideally with a theoretical grounding) and the claims/analyses are tied in with this definition. Such a grounded and precise setup would help in analyzing future generation discriminators that may not simply fall into the two discrete groups defined in the paper (i.e. sampling over-sensitive and sampling-insensitive). The results have promise, so the authors are encouraged to take into consideration the reviewer discussions to produce a stronger future submission.